# VSF: Simple, Efficient, and Effective Negative Guidance in Few-Step Image Generation Models By Value Sign Flip

**Wenqi Guo** [*]**& Shan Du** [†]
Department of Computer Science, Mathematics, Physics and Statistics
University of British Columbia
Kelowna, BC V1V 1V7, Canada
`wg25r@student.ubc.ca,shan.du@ubc.ca`

## Abstract

We introduce Value Sign Flip (VSF), a simple and efficient method for incorporating negative prompt guidance in few-step (1-8 steps) diffusion and flow-matching image and video generation models. Unlike existing approaches such as classifier-free guidance (CFG), NASA, and NAG, VSF dynamically suppresses undesired content by flipping the sign of attention values from negative prompts. Our method requires only a small computational overhead and integrates effectively with MMDiT-style architectures such as Stable Diffusion 3.5 Turbo and Flux Schnell, as well as cross-attention-based models like Wan. We validate VSF on a proposed challenging dataset, NegGenBench, with complex prompt pairs. Experimental results on our proposed dataset show that VSF significantly improves negative prompt adherence (reaching 0.420 negative score for quality settings and 0.545 for strong settings) compared to prior methods in few-step models (scored 0.320-0.380 negative score) and even CFG in non-few-step models (scored 0.300 negative score), while maintaining competitive image quality and positive prompt adherence. Our method also suppressed a generate-then-edit pipeline, while also having a much faster runtime. Code, ComfyUI node, and dataset are available in `https://github.com/weathon/VSF/tree/main`.

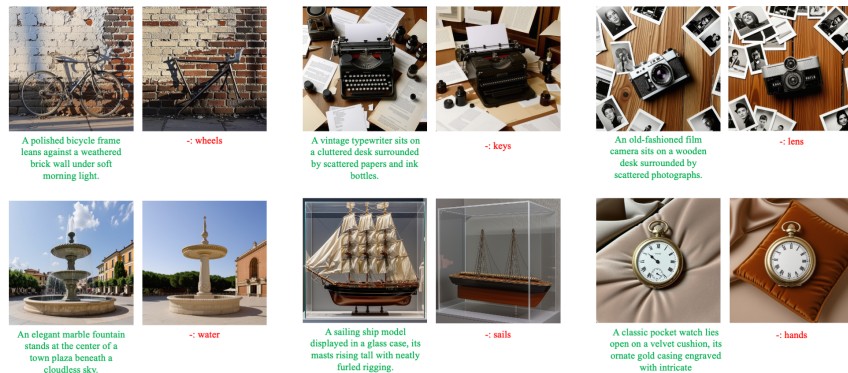

Figure 1: Original image without negative guidance and image generated using our VSF negative guidance on Stable Diffusion 3.5 Large Turbo. The green prompt is the positive prompt, and the red one is the negative prompt. These examples have significant challenges as they are removing essential parts of an object. The "hands" in the last image mean clock hands.

---

[*]Weathon Software
[†]Corresponding author

# 1 INTRODUCTION

Diffusion models (including flow matching models) have demonstrated their ability to produce diverse and high-quality images (Black Forest Lab, 2025; Woolf, 2022; Stability AI, 2024) and videos (Wan Team et al., 2025; Yin et al., 2025). However, a longstanding issue remains: the challenge of effectively applying negative guidance in image and video generation. Addressing this problem is crucial for improving content control, moderation (Schramowski et al., 2023), quality assurance, and reducing biases when generating general concepts (Chen et al., 2025a). However, vision language models (VLMs) have difficulties interpreting negations (Park et al., 2025; Alhamoud et al., 2025; Singh et al., 2025; 2024), rendering prompts containing negations ineffectively or made the negative prompt appears even more (e.g., a prompt like "a scientist who is not wearing glasses" will often generate a scientist with glasses—sometimes even more frequently than a simple prompt like "a scientist"). Classifier-free guidance (CFG) (Ho & Salimans, 2022) can be used to address this issue when substituting unconditional generation with negative guidance.

However, to enhance efficiency in image and video generation, numerous models have been distilled to support inference in just a few steps (1-8 steps), such as Flux Schnell (Black Forest Lab, 2025), Stable Diffusion 3.5 Large Turbo (Stability AI, 2024), SDXL Lighting (Lin et al., 2024), SNOOPI (Nguyen et al., 2024), and CausVid (Seppanen; Yin et al., 2025). However, CFG is incompatible with these models. These models are usually distilled and run in CFG-disabled mode, which means only the positive guidance is used, and there is no extrapolation. When CFG is applied forcefully, the resulting image often becomes oversaturated, particularly when the CFG scale is set high enough to suppress unwanted concepts. Moreover, if the number of diffusion steps is too low, the output may reflect features from both the positive and negative prompts (Nguyen et al., 2024), rather than excluding the negative prompt entirely. This occurs due to a divergence between the positive and negative guidance signals (Chen et al., 2025a). An example is shown in Figure 2. Additionally, even if CFG works, it requires two forward passes, one for positive guidance and one for negative guidance, which doubles the run time.

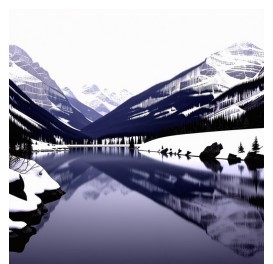

Figure 2: An example of forcefully applying CFG to a step-distilled model is shown using a guidance scale of 2.8 and only 4 steps on SD-3.5-Large Turbo. The positive prompt describes a Canadian winter landscape, while the negative prompt includes the word "lake." The resulting image does not cancel the lake and exhibits severe oversaturation artifacts (trees in the background).

To address this, two methods, Negative Steer Away Attention (NASA) (Nguyen et al., 2024) and Normalized Attention Guidance (NAG) (Chen et al., 2025a), have been introduced, employing negative guidance within attention final output space rather than the output space. NASA is currently limited to cross-attention models (though it can be re-implemented into other models), while NAG primarily targets quality control rather than negative prompt avoidance. Both methods calculate positive and negative attentions separately and subtract them using a prefixed scale (same as CFG), resulting in a fixed guidance strength throughout the generation, across different areas of the image, and at different layers of the model. This approach lacks adaptability to various time steps, layers, or image regions, limiting effectiveness in negative prompt adherence compared to a more adaptive method (Schramowski et al., 2023; Koulischer et al., 2025; Ban et al., 2024).

In this study, we introduce Value Sign Flip (VSF), a method that dynamically adjusts the guidance strength by flipping the sign of negative prompt values *within* the attention calculation (i.e., not the attention output). This enables the model to steer away from negative concepts adaptively based on their current presence strength, similar to the approach of Koulischer et al. (2025). VSF has a small computational overhead and, when combined with few-step models, facilitates extremely fast image or video generation ($< 3$ seconds). Our contributions in this work are: (1) we proposed a new method for better negative guidance; (2) we constructed a dataset, NegGenBench, consisting of challenging positive-negative prompt pairs; (3) we collected images generated using these three methods (VSF, NAG, NASA) and labeled their negative and quality score. We further fine-tuned a VLM on it for future work to better evaluate negative prompt following.

## 2 RELATED WORK

### 2.1 CLASSIFIER FREE GUIDANCE

Vision language models struggle to understand negation (Yuksekgonul et al., 2023; Singh et al., 2024; Alhamoud et al., 2025; Park et al., 2025) (We discussed more about this in the Appendix). Original classifier-free guidance (CFG) (Ho & Salimans, 2022) generates a conditioned noise prediction and an unconditioned noise prediction. In flow matching (Lipman et al., 2023), the predicted targets are the velocity ($u_t$). Thus, the original flow matching CFG prediction can be written as

$$u_t = f(\emptyset, x_{t+1}, t) + \lambda(f(p^+, x_{t+1}, t) - f(\emptyset, x_{t+1}, t)), \tag{1}$$

where $p^+$ is the positive prompt, $x_t$ is the latent at time $t$ (where higher $t$ means more toward the noise distribution), $f(\cdot)$ is the trained model, and $\lambda$ is the guidance scale. Later, the community finds out that by replacing the unconditional generation with a negative prompt (e.g., description of an unwanted image), the model will avoid the prompt due to the negative sign. This is the common implementation of a negative prompt. This turns the above equation into

$$u_t = f(p^-, x_{t+1}, t) + \lambda(f(p^+, x_{t+1}, t) - f(p^-, x_{t+1}, t)), \tag{2}$$

where $p^-$ is the negative prompt.

### 2.2 RECENT WORKS ON DYNAMIC NEGATIVE GUIDANCE

The studies on dynamic negative guidance are very limited (only (Ban et al., 2024; Koulischer et al., 2025; Schramowski et al., 2023)). Ban *et al.* (Ban et al., 2024) found that the negative prompts affect the model by delayed effects and neutralization. After the model has generated unwanted contents, the negative guided output ($u_{p^-}$) will neutralize the content. They also observed the reverse activation effect, where the negative prompt introduced early in the diffusion processes could actually induce the unwanted concepts. To address this, they proposed applying the negative guidance later in the diffusion process and found it effective.

Schramowski et al. (2023) used a very similar idea as CFG to avoid unwanted (NSFW) content. They generated an unsafe vector and purposely avoided it by subtracting it from the predicted noise. They also added a pixel-level guidance scale that depends on the pixel-wise distance between the positive predicted noise and the unwanted noise, making it adaptive to different regions in the image.

Koulischer et al. (2025) used similar ideas of both and proposed a temporal dynamic guidance scale method. They calculated a probability that the generated concept contains negative content and adjusted the guidance scale accordingly. However, their adaptive scale only changes throughout the steps and does not adapt to different regions in the image.

### 2.3 FEW-STEP IMAGE GENERATION MODELS

Traditional diffusion or flow-matching image generation models typically require many inference steps. However, with improved sampler, this can be reduced to around 20 steps. Recent approaches go further by using step distillation to reduce the number of steps to fewer than 8, or even a single step, as demonstrated in Flux Schnell (Black Forest Lab, 2025), SDXL Lightning (Lin et al., 2024), CausVid (Seppanen; Yin et al., 2025), SNOOPI (Nguyen et al., 2024), and Stable Diffusion 3.5 Turbo (Stability AI, 2024). Since these models are distilled, they generally do not use classifier-free guidance (CFG) during inference; when CFG is forcibly applied, the results are significantly degraded to the point that it is completely unusable (Nguyen et al., 2024), see Figure 2 for an example.

### 2.4 RECENT WORKS ON NEGATIVE GUIDANCE IN FEW-STEP MODELS

Recently, two approaches have specifically targeted negative guidance techniques for few-shot models: Negative-Away Steer Attention (NASA) (Nguyen et al., 2024) and Normalized Attention Guidance (NAG) (Chen et al., 2025a). Although they both focused on avoiding unwanted content and improving quality (using a negative prompt that describes bad quality), NASA mainly focused on avoiding unwanted content, while NAG focused on improving quality.

The authors of the NASA study found that neither standard CFG nor CFG applied directly to text embeddings yields desirable results in few-step scenarios, particularly in single-step settings. Specifically, the regular CFG independently computes positive and negative guidance signals, preventing the negative guidance from effectively neutralizing unwanted concepts. As a result, the produced images merely appear as a mixture of both positive and negative prompts unnaturally (an average image of the positive prompt generated image and the negative prompt independently generated image) rather than excluding negative prompt elements. Furthermore, the authors noted that applying CFG to text embeddings produces minimal benefits. For detailed examples and further illustration, readers could refer to the original paper introducing NASA (Nguyen et al., 2024).

NASA applies the guidance in intermediate states (attention outputs) instead of the final predicted noise or velocity. Specifically, they calculate a positive attention output $Z^+$ and a negative attention output $Z^-$, and the final attention $Z^{NASA}$ is obtained by subtracting the two with a factor $\alpha$, as shown in Equation 3. The $\alpha$ value is usually between 0 and 1.

$$Z^{NASA} = Z^+ - \alpha Z^- \tag{3}$$

Normalized Attention Guidance (NAG) used a similar approach. But instead of subtracting the negative attention map from the positive, it uses a similar extrapolation approach as CFG, as shown in Equation 4. The starting point $Z^+$ could also be replaced with $Z^-$; they are equivalent if $\phi$ is increased by 1.

$$\widetilde{Z}^{NAG} = Z^+ + \phi(Z^+ - Z^-) \tag{4}$$

However, to maintain the stability of the attention output space, they also applied normalization to $\widetilde{Z}^{NAG}$ to limit its norm relative to $Z^+$ with scale $\tau$ per token, resulting in $\hat{Z}$. Then it used a blending factor $\alpha$ to blend it with the positive attention result, as shown in Equation 5.

$$Z^{NAG} = \alpha \hat{Z} + (1 - \alpha)Z^+ \tag{5}$$

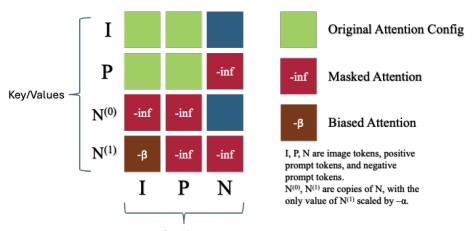

Figure 3: The attention mechanism of our method. We pass in image tokens ($I$), positive prompt tokens ($P$), and negative prompt tokens ($N$) into attention. For key and values, $N$ is duplicated, with values of one copy ($N^{(1)}$) scaled by $-\alpha$. Some areas are masked to avoid interference. An bias $-\beta$ is added to $I \rightarrow N^{(1)}$ attention.

The normalization and blending ensure the attention output of the NAG does not drift away from what the model usually sees during training, improving the quality of generated images. However, if the constraint is set to be too tight (i.e., high $\alpha$ and low $\tau$), it might also limit the model's ability to follow negative prompts.

### 2.5 OTHER RELATED WORKS

There is previous work that controls attention to manipulate images. Attend-and-Excite (Chefer et al., 2023) forces the attention on all key tokens to avoid the generation missing some terms in the prompt. Self-Guidance (Epstein et al., 2023) mantluplates attention to changing elements' properties in the image. BoxDiff (Xie et al., 2023) controls the attention map such that the object appears on the desired location in the image. However, our work is different in that we used it in a way to cancel unwanted elements by simply flipping the sign of values in the negative prompt.

Our work is also related to debiasing in image generation, since it can be used as a way to move away from learned associations (which could be about demography bias or object pairings). Previous work mainly targeted on the input, such as using reference images as input (Zhang et al., 2023) or using a learned prompt. FairQueue (Teo et al., 2024). Our method is different such that we control the value in the negative prompt. Additionally, our method can work against very strong associations like a bike without wheels.

### 3 PROPOSED METHODS

Our proposed method is built on top of NASA (Nguyen et al., 2024), Koulischer et al. (2025), and Schramowski et al. (2023). NASA has a fixed guidance for every attention calculation, Koulischer

et al. (2025) does not have token-level modulation, and Schramowski et al. (2023)'s approach of taking the differences between the positive and negative noise predictions is too simple for complex situations, as mentioned in Koulischer et al. (2025). In NAG's (Chen et al., 2025a) future work section, they also mentioned the possibility of token-level modulation but they did not propose a specific solution.

## 3.1 VALUE SIGN FLIP ADAPTIVE ATTENTION

We propose to expand Koulischer et al. (2025) idea to token-level modulation in few-step models. Let $W$ be a per-token weight at each attention calculation for how strongly the token is associated with a positive concept compared to the negative one. We can modify the NASA attention to Eq. 7. $W$ is obtained by a function with a positive prompt, a negative prompt, and an image as input.

$$W = g(p^+, p^-, I), \tag{6}$$

then we can rewrite the equation in NASA as

$$Z^W = WZ^+ - \alpha(1 - W)Z^- \tag{7}$$

An intuitive method to calculate $W$ involves using the model's attention map: when the image attends more to the negative prompt compared to the positive one, it should be steered away strongly accordingly. Thus, we can calculate the attention map between the image and the positive tokens $A^+$ and the image and the negative tokens $A^-$ before softmax calculation, then calculate their ratios to their sum. $Q$ is the image query tokens and $K^+$ and $K^-$ are the positive and negative prompt keys.

$$A^+ = \exp(\frac{Q(K^+)^T}{\sqrt{d}}), A^- = \exp(\frac{Q(K^-)^T}{\sqrt{d}}), W = \frac{\sum A^+}{\sum(A^+ + A^-)} \tag{8}$$

This approach involves complex attention calculation and two attention passes, but it can be implemented by a simpler approach. We can concatenate the values and keys of the positive and negative prompts, then flip the sign of the negative prompt values. This enables that when the image attends to the negative prompt, the flipped value of the negative prompt can cancel the unwanted content. The equation of our method in cross attention models, written in the matrix calculation, is shown in Equation 9, where $\oplus$ means matrix concatenation on the sequence length dimension, $\sigma$ is the softmax function on the sequence length dimension, and $V^+$ and $V^-$ are the positive and negative prompt values.

$$Z^{VSF} = \sigma(\frac{Q(K^+ \oplus K^-)^T}{\sqrt{d}})(V^+ \oplus -\alpha V^-) \tag{9}$$

This is similar to noise-canceling headphones, where a "flipped" wave is played to cancel the noise. Note that the key of the negative prompt is not flipped to keep the original meaning of the unwanted concept to match image patches. Mathematically, this is equivalent to $Z^W$. Proof is in the Appendix.

This approach gives a dynamic weight for the positive and negative prompts, and it varies for different layers, steps, and tokens.

## 3.2 ATTENTION MASKING AND DUPLICATION OF NEGATIVE EMBEDDING

The above method works well for cross-attention-based methods, where attention only exists between image-to-image in self-attention layers and image-to-text in cross-attention layers. However, it requires modification, including masking and duplication, to work in MMDiT-style models such as SD3.5 (Stability AI, 2024), where all image and text tokens are concatenated into a single sequence before attention.

In the standard MMDiT-style setup without our method (e.g., using CFG, NASA, or NAG), the sequence inputs for the attention module are: $[\mathbf{I},\ \mathbf{P}]$ and $[\mathbf{I},\ \mathbf{N}]$. If we concatenate all tokens into a single sequence without any modification, we will get: $[\mathbf{I},\ \mathbf{P},\ \mathbf{N}]$, where $\mathbf{I}$ represents image tokens,

$\mathbf{P}$ represents positive prompt tokens, and $\mathbf{N}$ is the negative prompt. During attention, queries, keys, and values are all projected from this combined sequence.

If we apply a sign flip to the negative prompt values by scaling $V_N = V\mathbf{N}$ with $-\alpha$ (where $V$ is the value projection), this flipped content affects all attention paths involving $V_{\mathbf{N}}$. That includes not only the intended interaction between image and negative prompt $(\mathbf{I} \to \mathbf{N})$[1], but also undesired interactions such as positive-to-negative $(\mathbf{P} \to \mathbf{N})$ and negative-to-negative $(\mathbf{N} \to \mathbf{N})$ (in which the value will cancel itself). These unintended interactions can distort the behavior of the model since the flipped signal influences more than just the image.

To address this, we introduce a duplication of the negative prompt. One copy remains unflipped and unscaled, denoted by $\mathbf{N}^{(0)}$, and the value (and only value) of the other is flipped and scaled, denoted by $V_{\mathbf{N}^{(1)}} = -\alpha \cdot V_{\mathbf{N}^{(1)}}$. The sequence becomes: $[\mathbf{I}, \mathbf{P}, \mathbf{N}^{(0)}, \mathbf{N}^{(1)}]$, where $\mathbf{N}^{(1)}$ does not act as query in attention calculation.

Additionally, inspired by Wang et al. (2025), where blocking some attention directions could improve quality, we apply attention masks to isolate the effect of $\mathbf{N}^{(1)}$ to only $\mathbf{I}$. Specifically, $\mathbf{N}^{(0)}$ is only allowed to attend to $\mathbf{I}$ and to itself, while $\mathbf{N}^{(1)}$ is only attended to by $\mathbf{I}$. Figure 3 shows the attention mask. Since $\mathbf{N}^{(1)}$ does not act as a key or value in any attention query, it doesn't produce associated output. Instead, $\mathbf{N}^{(0)}$ serves as the negative prompt tokens passed to the subsequent MLP layer and into the next attention layer, where it will be flipped again. It acts as an information collector from images to collect unwanted elements and also keeps updating itself from attention to itself, matching the prompt updating in a positive prompt.

This setup allows updates to the negative prompt based on attention from the image and from itself, and keeps the unflipped form active in the MLP path. It also prevents interference between positive and negative prompts and ensures that the flipped negative content affects only the intended image-to-negative attention path.

To preserve the high quality of generated images, we also applied attention bias $(-\beta)$ to $\mathbf{I} \to \mathbf{N}^{(1)}$ (also shown in Figure 3) and we removed the padding tokens from the negative prompt. Details and pseudo-code of our method are in the Appendix.

In our implementation, negative embeddings are duplicated once due to an implementation detail; however, this has a negligible impact on the final results after offsetting the scaling factor with 1 [2].

## 4 EXPERIMENTS

### 4.1 DATASET

Following Park et al. (2025), we use ChatGPT o3 (Open AI) to generate pairs of prompts and negative prompts to construct our dataset NegGenBench. Unlike prior work, our prompts are intentionally more challenging: the negative prompt is typically related to the positive one, and as a critical component—e.g., the positive prompt of a bike could have a negative prompt of "wheels". However, the positive prompt sometimes also uses a non-negation method to imply the item is missing, such as using terms like "empty" and "exposed" to make it more natural. Besides prompts, two questions are generated at the same time for later evaluation, one question asking if the image has the main object, either with or without the negative element, and the other one queries if the negative prompt element is missing. Prompts are generated in batches. There are 200 prompts generated, and we run them with 2 different seeds for the main results.

### 4.2 BASELINE AND METRICS

We chose NAG (Chen et al., 2025a) and NASA (Nguyen et al., 2024) as our baseline for few-step models. We also used a base model without negative guidance as a vanilla baseline, aiming to show the lower bound of the dataset. (i.e., how likely the positive prompt alone will help avoid negative concepts, if there is no negative guidance. This could happen either because the model is following

---

[1]The arrow direction is the attention direction, or the opposite direction of the information flow

[2]https://github.com/weathon/VSF/issues/19

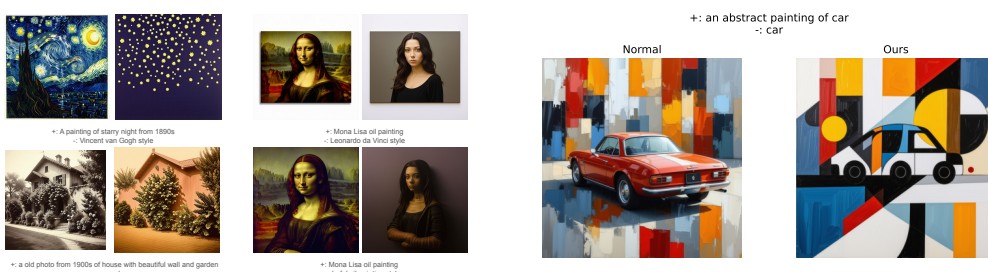

Figure 4: (Left) Style Avoidance Tests, (Right) (Semi)-abstract art generated by mentioning the main object "car" in a negative prompt. The car is semi-canceled and thus still present but in an abstract form.

the implication in the positive prompt, such as the word 'missing', or simply by chance.) Because NASA's original source code was not publicly available at the time of writing, we reimplemented it based on NAG's codebase. Specifically, we replaced the guidance equation from NAG (Eq. 4) with NASA's equation (Eq. 3), removed normalization and blending, and enabled guidance when the scale is greater than 0 (instead of 1). Additionally, we compared our method in non-few-step models with CFG and used other models as external baselines (External baseline results are in Table 1, but the experiment details are in the Appendix). We included Flux Kontext Labs et al. (2025) as an image-editing baseline. We first generate an image using SD-3.5-Large-Turbo, and then edit the image with Flux Kontext using the prompt `Remove [negative prompt]`. Since NAG was focused on quality instead of negative prompt avoidance, we re-tuned its hyperparameter such that it has stronger negation following in trade-off of quality and positive prompt following. We name this variance as NAG Strong. Same for our VSF method, we provided two different variations with different hyperparameters, focusing on quality (VSF Quality) and negative prompt following (VSF Strong). Hyperparameter details are in the Appendix.

Following Park et al. (2025); Wei et al. (2025), we used multimodal large language models (MLLM), specifically `llama-4-maverick-17b-128e-instruct-fp8`, to evaluate if the generated image follows the positive prompt and the negative prompt using the two questions generated during prompt generation. We did not use previous negation-aware CLIP-based work because they do not focus on missing an essential component, but simple meaning (e.g., a dog that is not on the grass). We did not use HPSv2 (Wu et al., 2023) or ImageReward (Xu et al., 2023) because they might give a low quality score for unusual objects (essential part being removed). Instead, MLLM is used to rate the image quality at the same time. At the end of our experiment, we also fine-tuned a Qwen-2.5-VL (Bai et al., 2025) model using data we generated by VSF, NAG, and NASA for better negation understanding. More details about the metrics and comparison with human validation are in the Appendix.

## 5 RESULTS

Quantitative results from using LLaMA as a judge evaluation are shown in Table 2, and qualitative results are shown in Figure 12 in the Appendix. Human validation is shown in Table 3 [3]. Automatic evaluation using the better negation-aware MLLM Qwen-2.5-VL is shown in Table 8 in the Appendix. Both the human validation and Qwen-2.5-VL results are aligned with our LLaMA evaluation relative ranking. It is important to highlight that the LLaMA assigns relatively generous quality scores; a score lower than 90 usually means the image already has degraded quality. Examples are in the Appendix Figure 11.

Based on the quantitative results, VSF Strong shows a significantly higher negative score than other methods, while maintaining comparable or better quality scores. Our more conservative method, VSF Quality, still achieved the second-highest negative score, with the highest quality score. Both VSF Strong and VSF Quality even achieve a higher negative score than traditional CFG in non-few-step models, demonstrating a stronger ability to avoid negative elements, even relative to the

---

[3]The results here might be negligibly different than our code due to implementation differences. `https://github.com/weathon/VSF/issues/19`

Table 1: External Baselines Comparsion

|  | Positive Score (↑) | Negative Score (↑) | Quality Score (↑) | ∼Runtime (↓) |
|---|---|---|---|---|
| *Open-weight Models* | | | | |
| VSF Strong | 0.870 | **0.545** | 0.952 | **3s** |
| VSF Quality | 0.980 | 0.420 | **0.986** | 3s |
| Generate+Edit | 0.875 | 0.488 | 0.958 | 55s |
| Janus-4o | 0.925 | 0.225 | 0.944 | 20s |
| Qwen-Image NP | 0.973 | 0.190 | 0.935 | 110s |
| Qwen-Image Negation | **0.990** | 0.100 | 0.937 | 110s |
| *Closed-weight Models* | | | | |
| GPT-4o | 0.978 | **0.705** | 0.954 | 47s |
| Nano Banana | **0.985** | 0.498 | **0.980** | **14s** |

established strong baseline. When compared with the external baseline, VSF also gets the highest performance in open source methods and only lags behind GPT-4o and achieves comparable performance with Nano Banana.

We also tested concepts/style avoidance, as shown in Figure 4 left. In the Starry Night example, VSF completely removed any signature elements in Vincent van Gogh's style, including the town, and generated a generic starry night image. Figure 4 also illustrates how our method can produce abstract art, which is typically discouraged during a model's finetuning since reward models favor realism. This is achieved through using the same word as the main object for both positive and negative scores in VSF, as detailed in the Appendix. VSF also has the ability to generate "anti-aesthetics" (unconventional, including abstract) art. Details of these results are provided in Figure 17 and Figure 18 (in the Appendix).

# 6 DISCUSSION

## 6.1 TRADE OFF CURVE

To systematically evaluate how effectively each model balanced positive prompt adherence, negative prompt adherence, and image quality, we conducted a hyperparameter sweep across each model. Specifically, we performed 66 runs for VSF and 287 runs for NAG, and 10 runs for NASA, with respect to their hyperparameter counts (2 for VSF, 3 for NAG, and 1 for NASA). A random sweep was executed besides for NASA, on which a single variable "grid" search is used, and evaluations were conducted using LLaMA, following the same criteria as previously described. Due to the large volume of runs, we limited our evaluation to the first 100 prompts with a single generation seed, potentially resulting in minor differences from earlier outcomes. Results are shown in Figure 5 Left.

From both plots, we observe that as the negative score increases, NAG and NASA both exhibit a significantly steeper and earlier decline compared to VSF in both positive and quality scores. In terms of positive score, VSF maintains scores above 90 even when the negative score rises to approximately 60. Regarding image quality, VSF similarly retains scores above 90 until a negative score of around 60, after which quality declines. In contrast, NAG and NASA both experience a sharp and early decline, with their quality score rapidly dropping to nearly 60 even before the negative score reaches 50. Keep in mind that a quality score under 90 means the image is already degraded, and if an image is rated 60, it is usually completely distorted. See Figure 11 in the Appendix for example.

Additionally, VSF demonstrates a broader operational range in negative scores. When necessary, it can achieve negative scores exceeding 70 while still preserving acceptable positive prompt adherence and image quality. Conversely, NAG and NASA become unacceptable in quality at negative scores below 50, limiting their practical effectiveness.

## 6.2 ATTENTION MAPS

Since our proposed method performs adaptive steering based on a negative attention map, we visualize the attention maps generated during the diffusion process in Figure 6 in Appendix. Extracting the full attention maps is difficult because efficient implementations, such as FlashAttention, do not explicitly store these maps, and storing and computing them will require a large amount of mem-

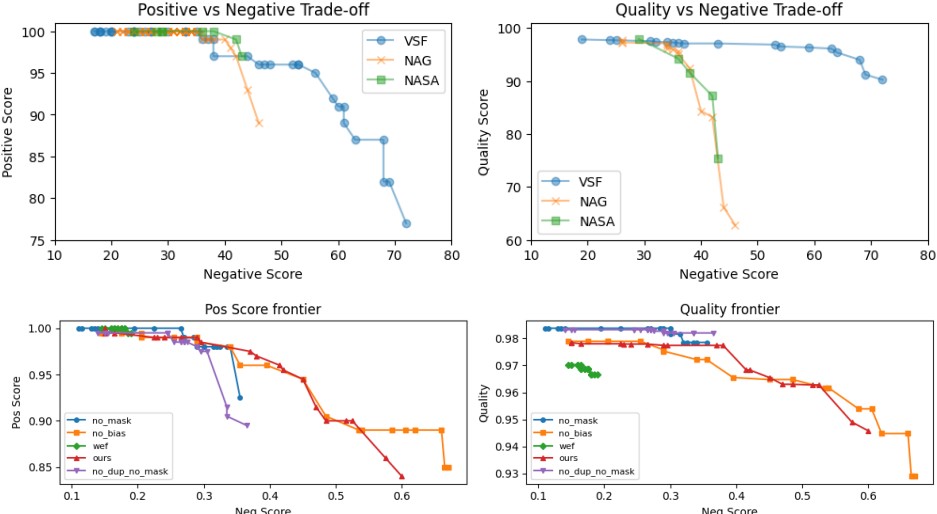

Figure 5: (Top) Trade-off plot of positive-negative score and quality-negative score. Both axies follows "higher is better." (Down) Trade-off plot of the ablation study.

ory. Therefore, we computed only the unnormalized attention values between the image tokens and negative prompt tokens. Figure 6 demonstrates that when the scale is set to 0, umbrellas appear, whereas setting the scale to 3 effectively avoids them. As indicated in the attention maps, image tokens corresponding to regions where umbrellas might exist (e.g., above human heads) exhibit higher attention toward the negative prompt tokens. Specifically, in steps 4 and 5, regions above the individuals on the left and right show strong negative attention, aligning with areas visually identified as umbrellas in $\alpha = 0$ image. In the final image, these highlighted regions no longer contain umbrellas, confirming that our method effectively suppresses the presence of undesired objects at specific locations.

## 6.3 ABLATION STUDY

To evaluate the effectiveness of each component of our approach, we conducted an ablation study using the following settings. For each setting, we scanned across scales for all 200 prompts using the same seed. Similar to before, we plotted the trade-off curve for each setting.

Rather than altering the attention values, we explored a simpler and more intuitive approach: flipping the text embedding prior to input into the DiT (Whole Embedding Flip, WEF). This is similar to applying the CFG on text embeddings studied in Nguyen et al. (2024), but keeps the positive and negative tokens separated. Specifically, the negative text embedding is scaled by $-\alpha$, concatenated with the positive prompt embedding in the sequence length dimension, and used as the prompt embedding for the DiT. We did not remove the padding for the negative prompt, as we found out that removing it causes the negative prompts to have no effect at all.

We also tested our approach with no bias, no mask (but still duplication), and no duplication no mask. The trade-off plot is shown in Figure 5 Right. The simpler and more intuitive WEF approach appears to have almost no effect at all. We hypothesize that this is because it is similar to flipping both the key and the value, causing regions most similar to the flipped key (i.e., least similar to the original negative prompt) to be pushed away, rather than pushing away regions most similar to the original negative prompt (i.e., unflipped key). From the figure, we can see that the configurations without masking have a sharp positive score drop as the negative score increases. The WEF has a very limited range of negative scores. Our methods and the one without attention bias have similar results; this could be due to the MLLM not being sensitive enough to see the minor changes in quality.

Ablation study on hyperparameters is shown in the Appendix.

Table 2: Positive scores (how well the model follows the positive prompts) and negative scores (how well the model avoids the negative prompts) of our model (VSF), NAG (Chen et al., 2025a), and NAG with hyperparameter re-tuned (NAG Strong).

|  | Positive Score (↑) | Negative Score (↑) | Quality Score (↑) |
|---|---|---|---|
| VSF Strong | 0.870 | **0.545** | 0.952 |
| VSF Quality | 0.980 | 0.420 | **0.986** |
| NAG (Chen et al., 2025a) | 0.993 | 0.220 | 0.968 |
| NAG Strong | 0.975 | 0.320 | 0.901 |
| NASA(Nguyen et al., 2024) | 0.970 | 0.380 | 0.867 |
| None | 0.990 | 0.195 | 0.968 |
| CFG (Ho & Salimans, 2022) (28 steps) | **1.000** | 0.300 | 0.956 |

Table 3: Human Labelled Metric For 10 Selected Prompts with 2 Seeds

|  | Positive Score (↑) | Negative Score (↑) | Quality Score (↑) |
|---|---|---|---|
| NAG Strong | 0.950 | 0.250 | 0.675 |
| NAG | **1.000** | 0.100 | **0.895** |
| NASA | 0.950 | 0.150 | 0.685 |
| VSF Quality | 0.900 | **0.550** | 0.823 |

## 7 CONCLUSION

In this paper, we introduced VSF, a novel approach for enhancing negative prompt adherence in image and video generation models. Our method involves flipping the sign of attention values and duplicating negative prompts and attention masking, effectively suppressing unwanted content. Experimental results indicate that VSF significantly outperforms previous methods, NAG (Chen et al., 2025a), NASA (Nguyen et al., 2024), and even CFG in terms of negative prompt adherence, with much lower trade-offs in overall quality and positive prompt following. We also showed that VSF can be applied to create more creative (by style avoidance, abstract images, and anti-aesthetics styles) images. VSF also only has one main hyperparameter and one minor hyperparameter, making it easier to tune them in downstream tasks. Future work directions are discussed in the Appendix.

## ACKNOWLEDGEMENTS

This work was supported by the NFRF under grant GR024801 and the CFI under grant GR024473. The authors acknowledge Weathon Software (`https://weasoft.com`) and Lambda, Inc. (`https://lambda.ai/`) for providing computing resources via Google Colab and Lambda Cloud, respectively.

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
