# Appendix

## Table of Contents

APPENDIX

## A  NEGATION IN VISION LANGUAGE MODELS

Much previous work has shown that existing vision language models (VLM) struggle to understand negation (Yuksekgonul et al., 2023; Singh et al., 2024; Alhamoud et al., 2025; Park et al., 2025). In classification tasks, the model cannot correctly understand text with negation in it, e.g. "a dog running" vs "a dog not running" might have very close embeddings, even though they are opposite. In our test using a CLIP-ViT-Base-32, the cosine similarity between "a dog running" and "a dog not running" is 0.9243, where the similarity between "a dog" and "a dog running" is only 0.8710. In Figure 7, we show 4 prompts "a photo of a bike", "a photo of a bike without wheels", "a photo of a bike with wheels", and "a photo of a car with wheels". We can see that the bike with wheels and the bike without wheels have extremely close embeddings. This problem has been introduced into text-to-image generation tasks, making it hard for the model to generate images without certain concepts

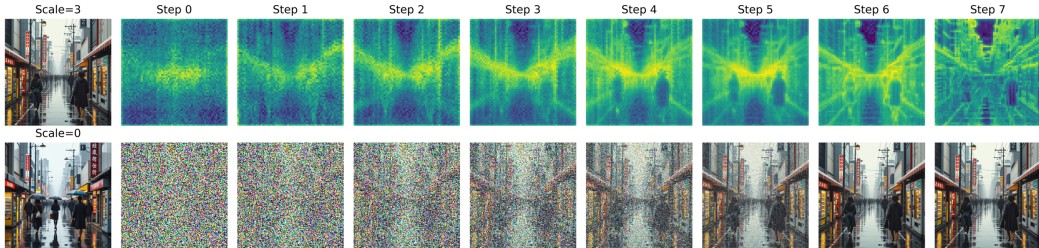

Figure 6: Attention maps and intermediate images during the diffusion process. The leftmost column shows the final generated image (top) and an image generated without applying VSF scaling ($\alpha = 0$, bottom). The top row on the right side displays the unnormalized attention values between image tokens and negative prompt tokens, while the bottom row shows the corresponding intermediate images at each timestep. The negative prompt is "umbrella."

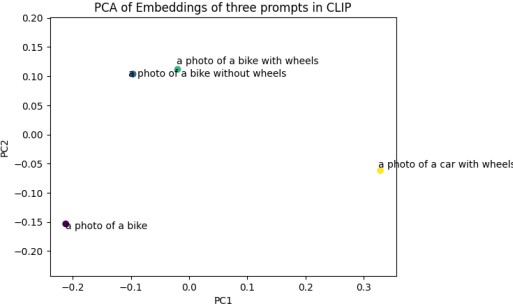

Figure 7: PCA plot of the 3 different prompts with a negation prompt.

(examples in Figure 1 of Singh et al. (2025) and Figure 5 of Park et al. (2025)). Thus, classifier-free guidance (CFG) was used to introduce a negative prompt to the image generation process. More details in the next section. Several studies have attempted to tackle this issue by employing alternative training strategies, such as incorporating harder samples in the training data designed for negation tasks (Yuksekgonul et al., 2023; Singh et al., 2024; 2025; Alhamoud et al., 2025; Park et al., 2025). Some of these methods have shown improvements in image generation tasks. For instance, Park et al. (2025) reported gains in Neg Score—measuring whether the model retains the primary subject while correctly omitting the negated object—for both SD-1.4 and SDXL-1.0, by replacing the default CLIP encoder with their NegationCLIP on their dataset, without additional T2I training.

These methods generally require re-training the text encoder (usually a CLIP-like model) with contrastive learning, which poses challenges for models that do not use contrastively pre-trained encoders, such as T5 (Raffel et al., 2023) in Stable Diffusion 3 (Stability AI, 2024; Esser et al., 2024) and Flux (Black Forest Lab, 2025). Moreover, each model using a different text encoder would require a separate, dedicated adaptation. Additionally, even if the text encoder understands the negation, the diffusion model might still fail to avoid certain items because of their strong association.

## B    PROOF THAT OUR METHOD IS THE SAME AS TOKEN-WEIGHTED SUBTRACTION

In this section, we prove that our method $Z^{VSF}$ is equivalent to token-weighted subtraction, denoted $Z^W$.

*Proof.* We define

$$A^+ = \exp(\frac{Q(K^+)^T}{\sqrt{d}}), \quad A^- = \exp(\frac{Q(K^-)^T}{\sqrt{d}}).$$

Then

$$W = \frac{\sum A^+}{\sum A^+ + \sum A^-}.$$

Substituting into the expression for $Z^W$:

$$Z^W = W \cdot \sigma(Q(K^+)^T)V^+ - (1 - W) \cdot \alpha \cdot \sigma(Q(K^-)^T)V^-,$$

and using the softmax definitions

$$\sigma(Q(K^+)^T) = \frac{A^+}{\sum A^+}, \quad \sigma(Q(K^-)^T) = \frac{A^-}{\sum A^-},$$

we obtain

Canceling the sums:

$$Z^W = \frac{A^+}{\sum A^+ + \sum A^-}V^+ - \frac{\alpha A^-}{\sum A^+ + \sum A^-}V^-.$$

This matches

$$Z^{VSF} = \sigma(Q(K^+ \oplus K^-)^T)(V^+ \oplus -\alpha V^-),$$

since

$$\sigma(Q(K^+ \oplus K^-)^T) = \frac{A^+ \oplus A^-}{\sum A^+ + \sum A^-},$$

and therefore

$$Z^{VSF} = \frac{A^+}{\sum A^+ + \sum A^-}V^+ + \frac{A^-}{\sum A^+ + \sum A^-}(-\alpha V^-) = \frac{A^+}{\sum A^+ + \sum A^-}V^+ - \frac{\alpha A^-}{\sum A^+ + \sum A^-}V^-.$$

Hence, $Z^W = Z^{VSF}$. □

## C  ATTENTION BIAS AND PADDING REMOVAL

We observe that even when the scaling factor $\alpha = 0$, including the negative prompt in the sequence still sometimes reduces image quality. This could be because the negative prompt "distracts" the image tokens' attention from the image tokens or positive prompts. To mitigate this effect, we introduce a negative bias $-\beta$ into the attention $\mathbf{I} \to \mathbf{N}^{(1)}$, thereby reducing the influence of the negative prompt.

In most models from Huggingface Diffusers (von Platen et al.), padding tokens in the text input are typically not masked during attention. This is likely because the models have learned to ignore padding, and masking them would add unnecessary overhead (due to some attention implementations like FlashAttention-2 (Dao, 2023) that do not support arbitrary masking). However, when we invert the sign of the padding tokens, it degrades output quality significantly. This could be because, although these tokens carry no semantic meaning, the sign-flipping introduces unseen states into the attention mechanism. To mitigate this, we remove padding tokens from the negative prompt embeddings. For the positive prompt, we retain padding tokens, as they do not introduce novel tokens and can improve generation quality. This aligns with training conditions and may allow the model to use padding positions as registers for auxiliary information.

## D  DETAILS ABOUT THE METRICS

To evaluate the scores of the generated images, we used LLaMA 4 Maverick, which has a very high image reasoning MMMU (Yue et al., 2024) score, higher than Gemma 3 and even GPT-4o. We avoided using the same model (o3) for both evaluation and generation for cost control and to avoid bias within a model. We did not evaluate the quality of the generated images using models like ImageReward (Xu et al., 2023) or HPSv2 (Wu et al., 2023) as in NASA or NAG, as current quality

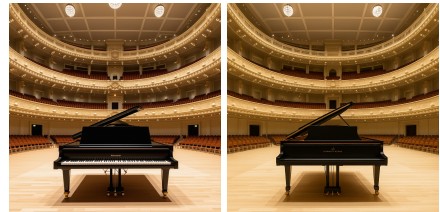

Figure 8: The left image is the original image, and the right image is generated by GPT-4o, where piano keys are removed. When scored using HPSv2, the left image got a score of 0.330 while the right image got a score of 0.319 using prompt of "A grand piano dominates an empty concert hall, a smooth ebony board stretching across the front." and the left get a score of 0.330 and the right got a score of 0.324 if we mention "no keys" in the promps.

or human preference assessment models do not account for negative prompts; traditional methods usually aim for real-world generations (Ye et al., 2025). Removing a key element from the positive prompt (e.g., removing the roof from a house) is likely to reduce perceived quality, since the result deviates from what is considered "normal," even though that is the intended outcome. An example is shown in Figure 8 where removing the keys from the piano results in a much lower quality score, even though other parts of the image look the same. Both ImageReward and HPSv2 are built on top of image-text alignment models (CLIP (Radford et al., 2021) or BLIP (Li et al., 2022)), which will likely lead to a decreased score when the main object is missing a critical part. Thus, we also let the MLLM rate the image quality from 0-1 for each image and told it to ignore the abnormality of following the negative prompt. We did observe that MLLM can make mistakes, especially when there is ambiguity or when the unwanted element is hard to see. However, we believe that in general, under 400 images, the mistakes are minor. To compensate for this, we also did a human evaluation and a more negotiation-aware MLLM evaluation to cross-validate the LLaMA evaluation. Due to model provider stability issues, we used different MLLM providers for different portions of the experiment for the same model under the same config; this could have some limited impact on the stability of metrics.

## E   HYPER-PARAMETER TUNNING

Although NAG (Chen et al., 2025a) also targeted negative concept avoidance, its primary focus was on its effects on improving generation quality (using words like "blurry" or "low quality" as a negative prompt). We believe the hyperparameters reported in their work were tuned with an emphasis on quality rather than negation handling. Therefore, we re-tuned their hyperparameters moderately and manually on guidance scale ($\phi$), blending factor ($\alpha$), and normalization factor ($\tau$). We will report experimental results on both original NAG (noted as NAG) and the improved hyperparameter version (noted as NAG Strong). The final hyperparameters used are $\phi = 11, \alpha = 0.5, \tau = 5$ for NAG Strong and $\phi = 4, \alpha = 0.125, \tau = 2.5$ for original NAG. This pushes the NAG to the edge of acceptable visual quality.

Similarly, for our VSF, we used two set of hyperparameters, VSF Quality ($\alpha = 3.3, \beta = 0.2$) maintained high quality and positive prompt alignment, while VSF Strong ($\alpha = 3.8, \beta = 0.2$) pushes it to the limit, reaching higher negative prompt alignment in trade of positive and quality score.

## F   HUMAN VALIDATION

To verify the results of the MLLM evaluation, we selected 10 prompts (with 2 seeds each) for VSF, NASA, NAG, and NAG Strong and manually labeled them with positive, negative, and quality scores. The human-labeled results are presented in Table 3. We validated MLLM performance by computing the binary F1 score and accuracy between MLLM outputs and human annotations. Cohen's kappa was not applied due to the highly imbalanced class distribution. The reliability metrics are summarized in Table 4. We observed that quality ratings from MLLM and human labels were uncorrelated in high-quality regions. To investigate this further, we evaluated quality scores

Listing 1: Pseudocode implementation of the Value Sign Flip (VSF) attention process

```python
# prep for embeddings
pos_embeds = get_embed(prompt)
neg_embeds = get_embed(neg_prompt, padding=False)
pos_len, neg_len, img_len = pos_embeds.shape[1], neg_embeds.shape[1], IMG_LEN

# concat positive and negative prompts (N0)
prompt_embeds = torch.cat([pos_embeds, neg_embeds], dim=1)

# prep for attention mask and bias (N1 never acts as query)
total_len = img_len + prompt_embeds.shape[1]
attn_mask = torch.zeros((1, total_len, total_len + neg_len))

# block P and N0 from attending to N1
attn_mask[:,-(pos_len+neg_len):,-neg_len:] = -torch.inf

# block image and P from attending to N0
attn_mask[:,:-neg_len,-(2*neg_len):-neg_len] = -torch.inf

# block N0 and N1 from attending to P
attn_mask[:,-neg_len:,img_len:img_len+pos_len] = -torch.inf

# bias image->N1 connections
attn_mask[:,:img_len,-neg_len:] -= offset

class VSFAttnProcessor(AttnProcessor):
    def __init__(self, attn_mask, neg_prompt_length):
        self.attn_mask = attn_mask
        self.neg_prompt_length = neg_prompt_length

    def forward(self, hidden_states, encoder_hidden_states, attention_mask):
        # get qkv projection for image tokens
        q = self.get_q(hidden_states)
        k = self.get_k(encoder_hidden_states)
        v = self.get_v(encoder_hidden_states)

        # get qkv projection for encoder tokens

        q_enc = self.get_q_encoder(encoder_hidden_states)
        k_enc = self.get_k_encoder(encoder_hidden_states)
        v_enc = self.get_v_encoder(encoder_hidden_states)

        query = torch.cat([q, q_enc], dim=2)

        # append P, N0 (in k_enc and v_enc) and N1 (the last portion of k_enc and v_enc) at the end
        k = torch.cat([k, k_enc, k_enc[:,:,-self.neg_prompt_length:]], dim=2)
        v = torch.cat([v, v_enc, v_enc[:,:,-self.neg_prompt_length:]], dim=2)

        # sign-flip values of N1
        v[:,:,-self.neg_prompt_length:] *= -scale

        hidden_states = F.scaled_dot_product_attention(
            query, k, v,
            dropout_p=0.0, is_causal=False,
            attn_mask=self.attn_mask.to(query.dtype)
        )
        hidden_states = hidden_states.transpose(1, 2).reshape(batch_size, -1, attn.heads * head_dim)
        return self.out_proj(hidden_states)

for block in model.transformer.blocks:
    block.attn1.processor = VSFAttnProcessor(attn_mask, neg_len)

# diffusion process continues
```

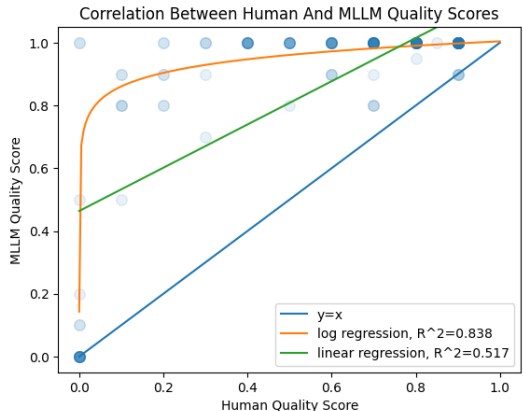

Figure 9: Correlation Between the Human-rated quality score and MLLM-rated quality score

Table 4: Reliability Metric For Human And MLLM Evaluation Results

|          | Negative Score | Positive Score |
|----------|----------------|----------------|
| F1       | 0.667          | 0.974          |
| Accuracy | 0.850          | 0.950          |

over a broader set of conditions. With a large sample size, we found that correlation emerges in a wider range: when scores are close to 1, small fluctuations carry little meaning, but substantially lower scores (e.g., $< 0.9$) may indicate degraded quality. The correlation is shown in Figure 9. This supports the observation in Figure 11, where MLLM tends to overestimate quality. From the scatter plot and regression, we can see that the MLLM score is usually higher than the human score, and although they are not linearly correlated, they are monotonically correlated.

## G   ADAPTING TO OTHER DiT MODELS

In this paper, we primarily use SD-3.5 (Stability AI, 2024) due to simplicity and elegant architecture. However, our method can theoretically be adapted to any transformer-based diffusion or flow-matching model. To demonstrate this adaptability, we implemented our method on Wan 2.1 with CausVid LoRA (Yin et al., 2025; Seppanen) and Flux Schnell (Black Forest Lab, 2025).

For Wan 2.1, which uses cross-attention between image and text, duplication and masking are unnecessary and not used. Because our approach does not perform extrapolation and solely provides negative guidance, it cannot enhance overall quality significantly or replace CFG sampling in non-disstilled models, making it incompatible with the original Wan 2.1 model. Instead, we utilize CausVid (Yin et al., 2025), which enables Wan to function effectively without classifier-free guidance in few-step settings. Specifically, we used a LoRA distilled from the original CausVid that can be directly applied on top of Wan 2.1 (Seppanen). For qualitative results from Wan, please see the appendix.

We also tested our method on Flux Schnell (Black Forest Lab, 2025). However, due to the model likely being trained to associate items with their associated items that often appear together strongly, we need to make some modifications. Before the negative prompt was fed into the model, we did a CFG-like extrapolation on the negative prompt, with a mean padding embedding as a null condition. This follows the implementation of the Compel package. Noted as:

$$p^- = p^- + \lambda \cdot (p^- - p^{\emptyset}), \tag{10}$$

we used $\lambda = 8$ in this case. Quantitative results are shown in Table 5. We can observe that even without any negative guidance, the Flux Schnell model can slightly better avoid the items solely based on the positive prompts (since the positive prompts implied the item is missing using terms like "empty"), but with the help of VSF, it further increases the negative score without compromising the positive and quality score.

Table 5: Comparsion of Flux Schnell VSF and Original Schnell (Black Forest Lab, 2025)

| Method | Positive Score | Negative Score | Quality Score |
|---|---|---|---|
| Flux Schnell | 1.00 | 0.22 | 0.99 |
| Flux Schnell VSF | 0.97 | 0.41 | 0.99 |

Table 6: The computation cost of each model. Time is measured in total runtime per sample, and VRAM is the peak RAM during the 25 samples generation. Since VSF Wan does not require a mask, and it is only used for bias, we also tested it without the bias. The SD3.5 model used is SD-3.5-Large-Turb,o and the Wan model used is Wan-2.1-T2V-1.3B.

| | Wan | | SD3.5 | |
|---|---|---|---|---|
| | Time | VRAM | Time | VRAM |
| Baseline | 23.10s | 22.05GB | 2.14s | 28.49GB |
| NASA | - | - | 2.89s | 28.50GB |
| NAG | 25.58s | 22.06GB | 2.98s | 28.50GB |
| CFG (Theroatical) | 46.20s | - | 4.28s | - |
| VSF | 22.70s | 23.05GB | 3.00s | 28.53GB |
| VSF (No mask/bias) | 22.70s | 22.05GB | - | - |

## H    COMPUTATIONAL COST

Since our method does not require two passes through the entire model (as in CFG) or the attention module (as in NAG or NASA), and only slightly increases the sequence length ($< 0.2\%$), its theoretical computational cost is significantly lower, close to that of a single pass. However, due to implementation limitations (specifically, FlashAttention-2's lack of support for arbitrary attention masking), the actual runtime of our method is higher than the original single-pass MM-DiT models, and similar to NAG or NASA, but still lower than CFG.

To accurately measure the computational cost, we evaluate the runtime of 25 identical prompts under four settings: no guidance, NAG, NASA, and our proposed guidance, VSF, and then report the average runtime and peak memory usage for each setting. We also reported the theoretical CFG time as double the one without guidance. To avoid GPU thermal throttling affecting the results, we pause for at least 5 minutes between each set of tests. The tests are done on NVIDIA A100 40GB on Google Colab, as this is the most accessible option for high-end GPUs for users. Stable-Diffusion-3.5-Large-Turbo is generated in 8 steps for 1024x1024 resolution, Wan is generated in 8 steps with 480x832 resolution, and 81 frames. The results are shown in Table 6.

From the table, VSF requires marginally more time and memory than NAG in SD3.5, while they are both significantly faster than theoretical CFG time, which would be twice the baseline. In Wan, VSF outperforms NAG and is even slightly better than the baseline (likely due to nature variation or noise) in terms of compute time, though it consumes 1GB more memory, likely due to the attention bias being stored. Since this bias is optional, we tested VSF Wan's performance with it removed, which results in an improvement in VRAM usage such that it uses the same amount of VRAM as baseline and NAG, and no change in runtime.

## I    EXTERNAL BASELINES

In addition to other guidance methods applied to SD-3.5-turbo, we evaluated several external baselines. The first baseline employs a generate-then-edit approach, loosely inspired by Generate-Plan-Edit (GraPE) (Goswami et al., 2025), but omitting the planning stage as our goal is straightforward (removing unwanted elements). Specifically, we first generated images using SD-3.5-Large-Turbo without a negative prompt, and subsequently edited out the unwanted elements automatically using Flux Knoest (Labs et al., 2025), an image editing model, using prompt `Remove [negative prompt].`

The second baseline utilizes GPT-4o's native image generation capability. GPT-4o has demonstrated strong prompt-following performance(Wei et al., 2025), including in handling negation tasks. As GPT-4o lacks explicit negative prompt functionality, we formatted prompts as `[Positive prompt], but with no [negative prompt].` Since our focus is on evaluating negation

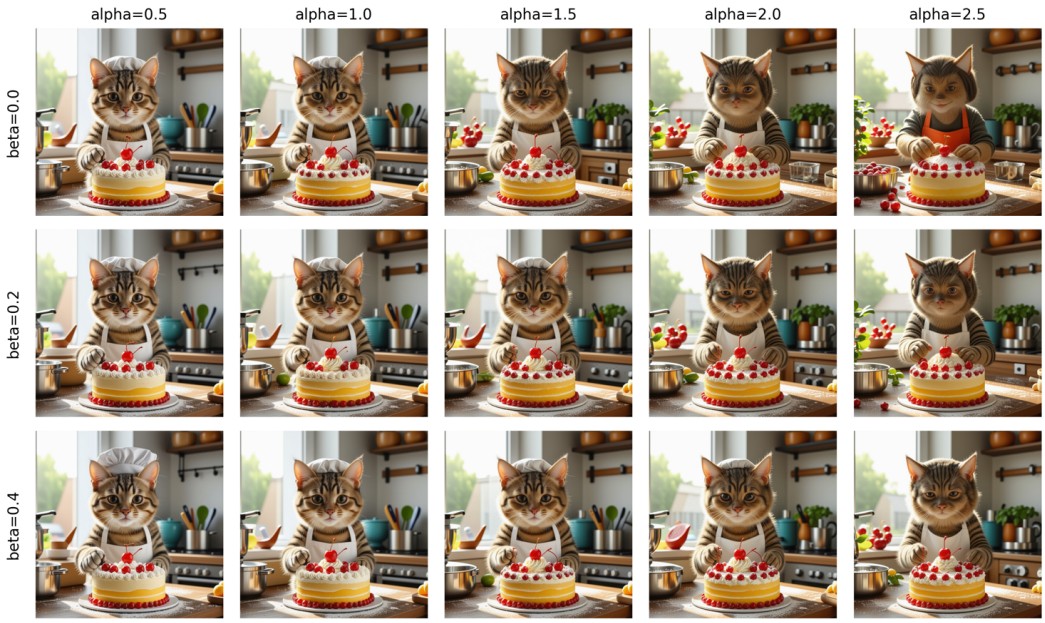

Figure 10: Effects of guidance scale ($\alpha$) and attention bias ($\beta$) in image generation. Positive prompt is "a cat making a cake in the kitchen, the cat is wearing a chef's apron..." and negative prompt is "chef hat."

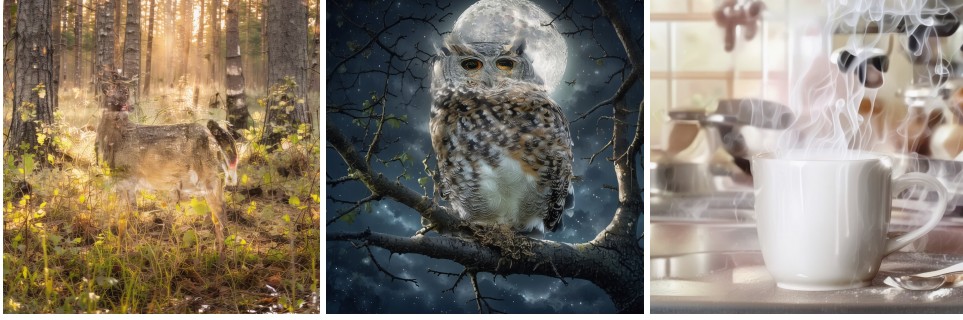

Figure 11: An example of a completely distorted image gets a relatively high quality score. The left one has a score of 70, the middle one has a score of 90, and the right one is a slightly distorted image, but still rated for 100.

rather than image quality, we adopted the "low" generation setting. Besides GPT-4o, we also added the newly released Nano Banana from Google. It is also a language model-based image generation model and has received a good reputation in the image generation community.

The third baseline we included is Janus-4o (Chen et al., 2025b), a model distilled from GPT-4o onto the Janus-Pro base architecture (Chen et al., 2025c). Given GPT-4o's strong prompt-following performance, we anticipated competitive results from Janus-4o. We provided negative prompts directly as negations within the positive prompts, same as GPT-4o.

Finally, we tested Qwen-Image (Wu et al., 2025) using two configurations: one employing separate positive-negative prompt pairs using CFG (labeled as Qwen-Image NP), and another embedding negative prompts as negations within the positive prompt itself (labeled as Qwen-Image Negation), while still using CFG with an empty negative prompt. Qwen is run under DFloat-11.

All measure time is measured on Google Colab 40GB A100 GPU, and for Qwen-Image and Generate+Edit, model CPU offloading is enabled.

The results are presented in Table 1 in the main text. The table indicates that VSF Strong achieves the second-best negative score, only behind GPT-4o, while also demonstrating a significantly faster runtime compared to all other methods, outperforming even the generate-then-edit pipeline. The GPT-4o distilled model, Janus-4o, has an unexpectedly low negative score, which could be because they did not have enough negation-included prompts in the distillation data. The VSF Quality had a lower negative score compared to Generate+Edit, while having a much higher positive and quality score, and shorter runtime.

## J  SD3.5-LARGE-TURBO QUALITATIVE RESULTS

Selected qualitative results are shown in Figure 12. The positive prompt is condensed for spacing. For the glasses without lens images, both NAG and NAG Strong generated classes clearly have a lens. For the VSF-generated image, we can see the lens is missing, even though the frames are floating. However, this issue was also presented in NAG Strong's image, even though it still has glasses. For a sailboat without sails, all other methods generated smaller but still existing sails, while VSF successfully avoided sails. In the third image of a lighthouse without a lamp, both VSF and NASA have no visible lamp, while the images from NAG and NAG Strong have a clear lit lamp. In the image of a bicycle without a chain, NASA generated a blurry image without bikes at all, while NAG generated a normal image, and NAG Strong generated a slightly distorted image of a bike with no seat yet the chain is still present. VSF successfully generated a bike without a chain, even though it also removed the seat. For the prompt of a lantern with no glass panes, NASA generated a lamp with frosted panes, NASA++ generated a classic glass pane, and NASA++ generated broken frosted panes. VSF, in this case, generated a pane that is clearly not glass. In the last example of a T-shirt, NASA generated a blurry image with still one sleeve visible, and NAG generated the T-shirt with both sleeves visible. NAG Strong and VSF both avoided the sleeve, even though NAG Strong has some artifacts.

## K  QUALITATIVE RESULTS FOR WAN

In Figure 13, we showed 3 examples generated from Wan-2.1-14B. In the first example, we successfully removed the stars in the background while keeping other elements intact. On the right side, in the absence of stars, the moon lander is generated to fill the space. In the second video, we successfully generated a windowsill without a curtain. In the last video, the generated video from VSF contains no trees on the left, and instead, it fills it with a hill. There are still some bushes on the right side, which do not violate the negative prompt of "trees. All videos have the same high quality as the original one.

## L  FAILURE CASES

Like any method, our method is not perfect, especially in a challenge dataset like NegGenBench. In Figure 14, we showed 4 failed cases. In the case where we want to generate a keyboard without

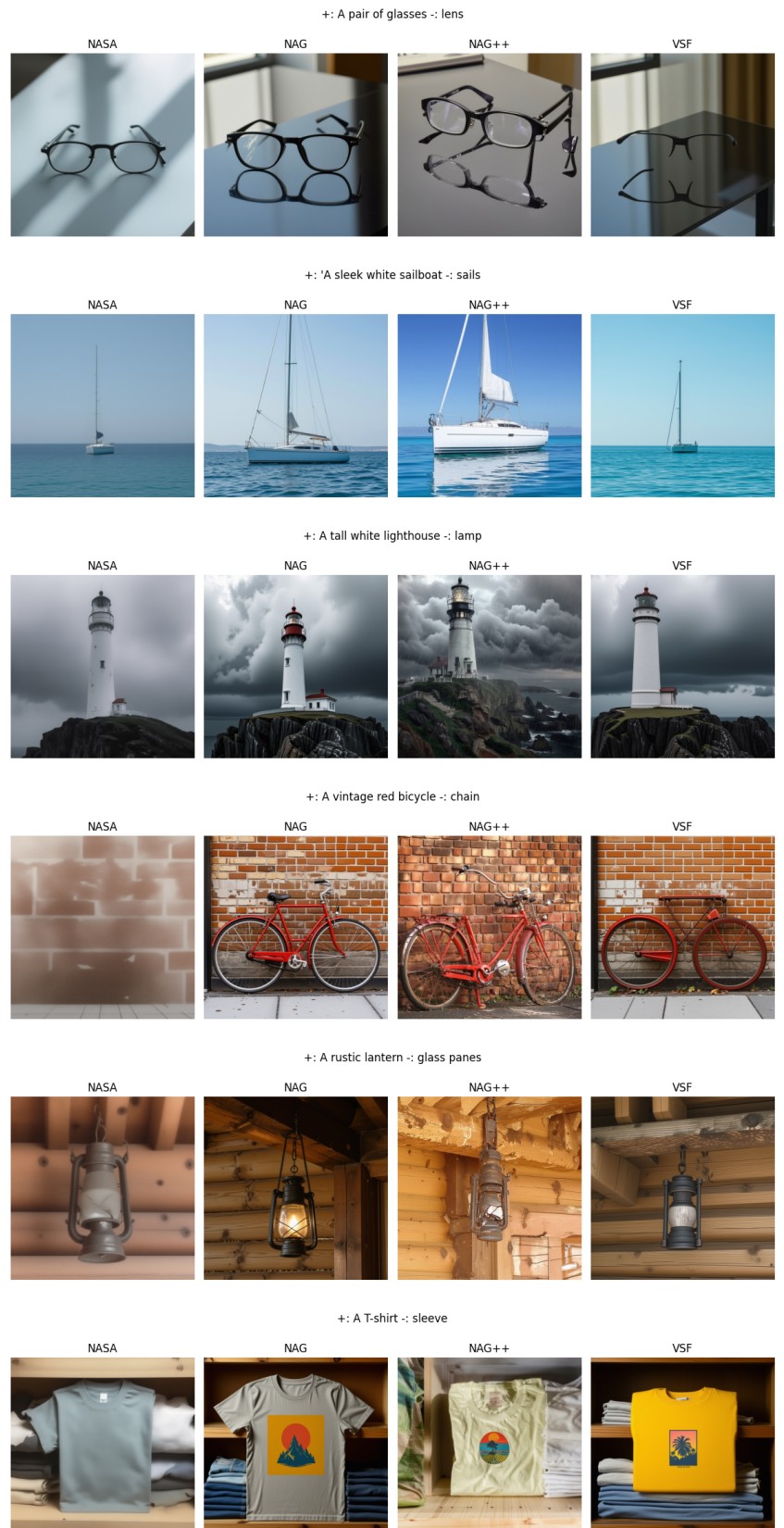

Figure 12: Selected Results for Comparison. Positive promtes are condensed for spacing.

**Positive Prompt:** An astronaut hatching from an egg, on the surface of the moon, the darkness and depth of space realised in the background. High quality, ultrarealistic detail and breath-taking movie-like camera shot.
**Negative Prompt:** stars in the sky, low quality, blurry, distorted, low resolution, unatural motion, unnatural lighting

Original                                VSF

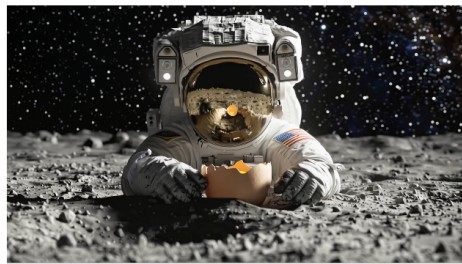 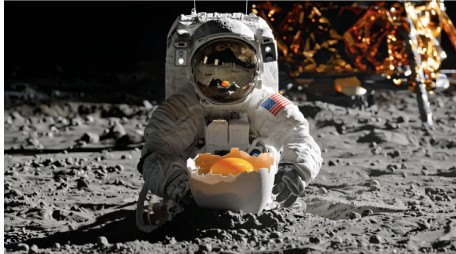

**Positive Prompt:** A short-haired gray cat sitting alert on a windowsill, its cheeks unusually smooth beneath attentive eyes.
**Negative Prompt:** curtain, low quality, blurry, distorted, low resolution, unatural motion, unnatural lighting

Original                                VSF

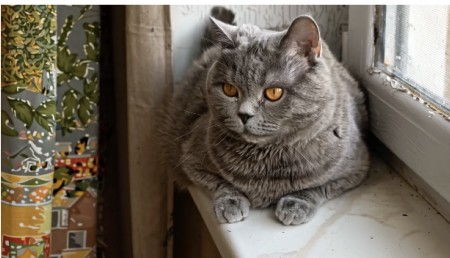 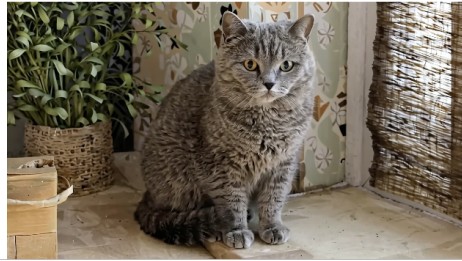

**Positive Prompt:** A mountain bike rides along a winding countryside trail under a cloudy night sky. The moon is clearly visible through gaps in the clouds, casting faint silver light over the uneven terrain. The bike's headlamp cuts through the darkness, illuminating the rocky path ahead as it speeds forward. The ambient sounds of distant winds and gravel crunching beneath tires accompany the scene.
**Negative Prompt:** tree, low quality, blurry, distorted, low resolution, unatural motion, unnatural lighting

Original                                VSF

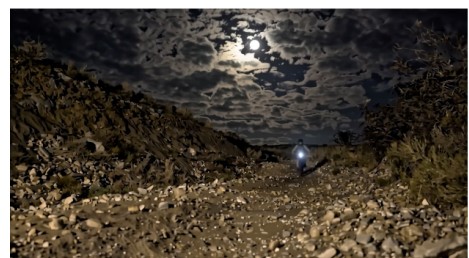

Figure 13: Qualitative Results for Wan

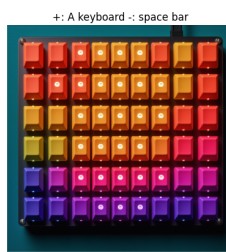 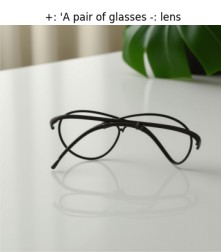 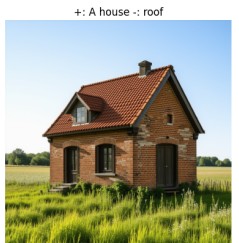 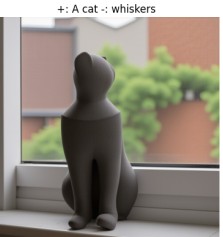

Figure 14: Failed examples, positive prompts are condensed for spacing.

Prompt: an abstract gouache drawing of car

Figure 15: Image with abstract style receives a lower score in HPSv2 (reflecting human preference; traditionally, models aim for higher scores).

a spacebar, the generated object is technically a key-board (an array of keys) and has no space bar, but it is not what people imagine when they think about "keyboard with no spacebar." The second failed case is another image from glasses with no lens; the generated image has no lens, but the frame is twisted in an unnatural way. In the third example, where a house with no roof is needed, VSF completely missed the negative prompt, possibly due to the strong association between roof and house. In the last example of a cat with no whiskers, the generated image technically has no whiskers and looks like a cat, but it looks more like a cat statue instead of a living cat.

## M  NON-OBJECT NEGATIVE GUIDANCE

In this paper, we focused on removing a critical component in the image. To further validate our negative guidance method in other areas, we also tested it on style avoidance. In Figure 4 (in main text), we show four examples, each of which is generated using the same seed. We can see that when prompted with famous artwork (e.g.,"A painting of Starry Night from the 1890s" or "Mona Lisa oil painting") but with a negative prompt of the artist's name style, the generated image avoided any elements related to the style (including the town in the Starry Night) but kept the semantic meaning of the positive prompt. When prompted to give an old photo but not monochrome, the generated image is more like an old-style color photo, follows both non-monochrome and also not very bright (as old photos, even in color, are less vibrant). We find these examples interesting and think they can be used for machine unlearning, using a similar method as in (Gandikota et al., 2023).

## N  AN EXPERIMENT ON ANTI-AESTHETICS ARTS

Current image generation models are typically finetuned to align with so-called "human preference." However, we argue that there is no universal standard for human preference, and it cannot be defined solely by developers, who inevitably bring their own interests and assumptions. Aligning models exclusively with such values risks introducing bias and potentially marginalizing minority perspectives and interests (Arzberger et al., 2024; Turchin, 2019; Sutrop, 2020; Guo et al., 2025).

In the context of image generation, this alignment may lead to homogenization of style or taste, producing only broadly pleasing outputs for the general population. Such uniformity can suppress niche demands for degraded, low-quality, or unconventional aesthetics. To counteract this, one possible approach is the use of negative guidance to steer outputs away from mainstream preferences. In this experiment, we tested how VSF can address this issue. We ran our VSF in settings where $\alpha = 0$, which shows on the left, and $\alpha \in [0, 4]$, which shows on the right side. The image with $\alpha = 0$ might not be the same as the one without guidance, but should be an image without negative guidance. The first test used prompts containing the same object in both positive and negative form, with the goal of producing abstract art. This works by semi-canceling the main object, making it appear in an abstract form. Abstract styles are often disadvantaged in alignment settings, since reward models typically favor realistic or figurative outputs. VisionReward (Xu et al., 2025) encodes this bias through its scoring metric, and LAPIS (Maerten et al., 2025) reports that abstract paintings generally

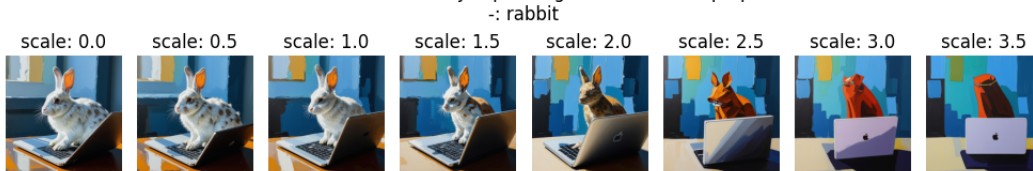

Figure 16: Abstraction of the image as scale increases.

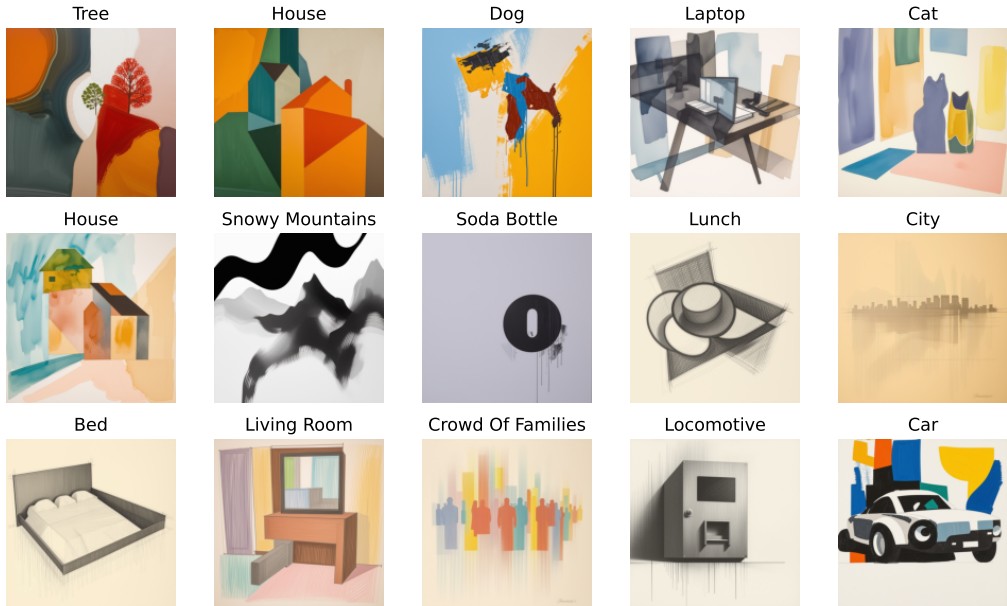

Figure 17: More Abstract Arts Examples

receive lower preference scores. Figure 15 shows that an abstract image gets a much lower score compared with a figurative one. As shown in the first two rows of Figure 18, the apple, people, and cat appear in abstract form, demonstrating a clear shift away from the default figurative tendency of the aligned models when VSF is applied. For the last image of a dog, we used "cute" as a negative prompt, which usually describes realistic objects, and we achieved a very abstract and artistic image. Figure 16 shows how abstract the image gets as the scale increases. More examples are shown in Figure 17.

In the second test, the goal was to diverge from styles that are generally appreciated. The positive prompt specified the desired style, while the negative prompt contained descriptions of commonly preferred styles. Importantly, the positive prompt clearly described the intended output, so a faithful model should follow it rather than default to generalized human preference. The tested cases included desaturated color, sad emotion, pixelated art, insufficient lighting, unnatural colors, and a non-beautiful cat. Results show that the baseline model struggled to maintain these characteristics, often reverting to conventionally "beautiful" imagery, whereas VSF successfully produced outputs aligned with the specified unconventional styles.

## O    NEGATION-AWARE MLLM

Upon visual inspection, we observed that GPT-4o effectively avoids many negative prompts, though occasionally the ambiguity within negative prompts (e.g., the term "door" referring either to the door panel or the entrance) and the vision ability of MLLM itself (i.e., hallucination) leads to lower negative scores. It is possible that the negative scores for other methods might also be underesti-

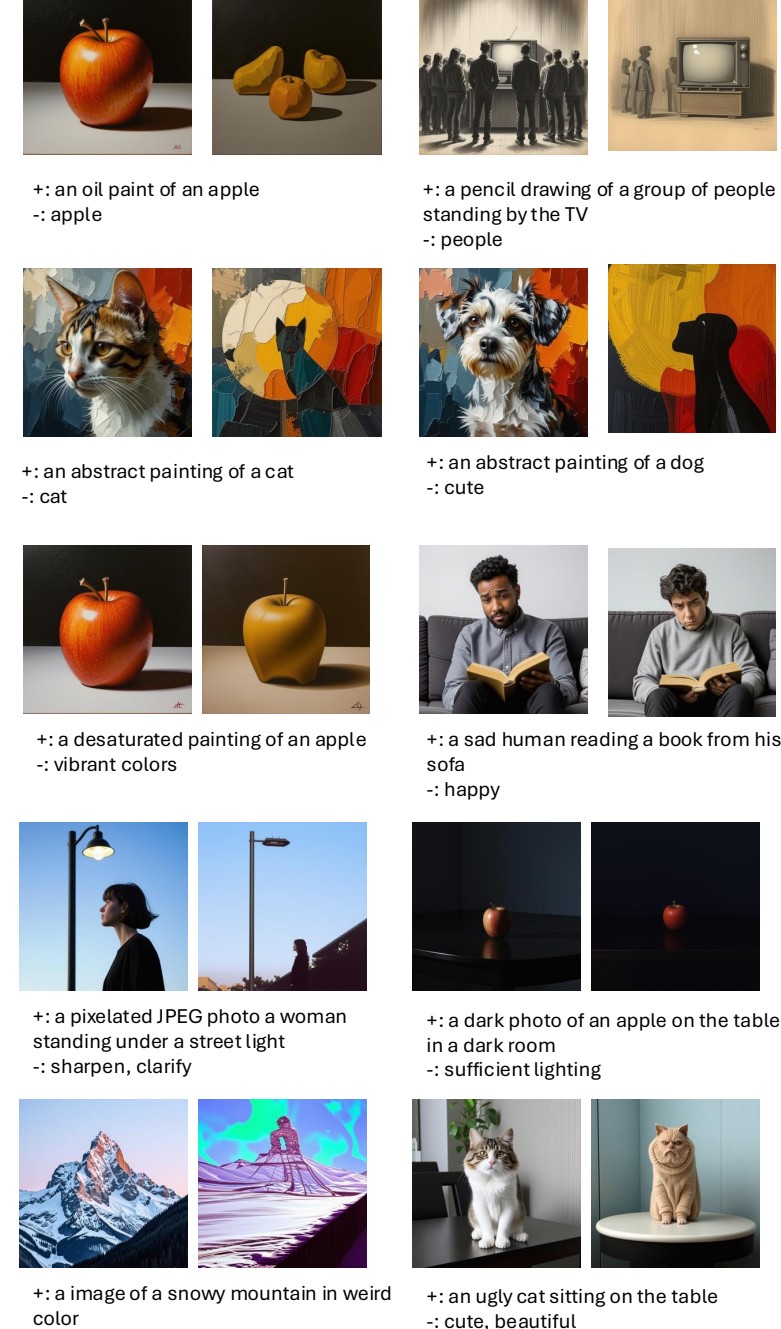

Figure 18: A test for anti-aesthetics. The left image is generated with $\alpha = 0$, and the right image is generated with $\alpha > 0$. These tests aim to move away from universally pleasing styles and demonstrate the ability to capture more diverse aesthetic preferences.

Table 7: Negation-Aware LLM Evaluation on Testing Set

|  | Parameters | r ($\uparrow$) | Acc ($\uparrow$) | F1 ($\uparrow$) |
|---|---|---|---|---|
| Llama Maverick | 400B | 0.05 | 0.83 | 0.59 |
| Llama Maverick CoT | 400B | 0.03 | 0.77 | 0.50 |
| Qwen-2.5-VL 32B | 32B | 0.28 | 0.80 | 0.36 |
| Qwen-2.5-VL 32B CoT | 32B | 0.31 | 0.88 | 0.70 |
| NegAwareQwen-7B | 7B | **0.37** | 0.86 | 0.65 |
| NegAwareQwen-32B | 32B | 0.34 | **0.90** | **0.76** |

mated. We acknowledge this as a limitation associated with using an MLLM as the evaluator. Thus, we provided a better negotiation understanding of MLLM and used this MLLM to evaluate different guidance methods.

To enable future research on evaluating negation prompting in generative models, a reliable and fast (LLaMA Maverick is too big) evaluation model is necessary. Previous CLIP-based studies concentrated on simple negations (e.g., "a cat that is not on the grass") rather than more complex cases such as those in NegGenBench. To address this, we finetuned a multimodal large language model, Qwen-2.5-VL-7B, called NegAwareQwen for improved understanding.

We created 100 additional prompts using GPT-5, each paired with a positive and a negative pair and 2 questions. For each prompt, we generated two images with the three models (NAG, VSF Quality, and NASA) and selected 722 for manual scoring on two dimensions: adherence to the negative prompt and overall quality. We did not assess the positive prompt evaluation as those are simpler, and almost all images in the dataset have a perfect positive prompt following. Negative adherence was rated on a three-level scale: 0 (ignored), 0.5 (partial), and 1 (fully followed). When generating the samples, we slightly randomly adjust the hyperparameter in a small range to create more diverse data (For VSF, $\alpha = 3.3 \pm 1, \beta = 0.2$; for NASA, $\alpha = 0.15 \pm 0.05$; for NAG, $\phi = 8 \pm 4, \alpha = 0.5 \pm 0.2, \tau = 4 \pm 2$). Since this makes image generation models generate suboptimal images, the rating results of each model's images are not used for direct comparison. The dataset and the model will be opened after publication.

Note that here Llama showed a very weak r-score for quality; this is because all the images are evaluated using relatively high-quality images (unlike in the ablation study, where many images are lower quality). We did not compare the 7B untrained model because it often failed to output the structure data needed.

The model was finetuned using prompts from the dataset of all 3 models. We trained the model using QLoRA (Dettmers et al., 2023) with rank of 8 and $r = 8$, applied to query, key, value projections in both the vision encoder and language model with dropout of 0.1. Model is trained using $lr = 5 \times 10^{-5}$ (with warm up and decay), WeightDecay=0.1, BatchSize=16, Epoch=5. The dataset is split into train-val with a 90-10 ratio based on the prompt level splitting and aiming for balanced scores in each split. We treat 0.5 as False and calculate negative scores as a binary metric. Results are presented in Table 7 with comparison with the same model without finetuning and LLaMA-Maverick.

## P  EVALUATION USING NEGAWAREQWEN

We re-evaluated VSF Quality, NAG, NAG Strong, NASA, GPT-4o, and Nano Banana images generated using the prompts and seeds in the main text of the paper using our finetuned NegAwareQwen; the results are shown in Table 8. We did not round the 0.5 score. We used a positive score from the original Table 2 as our finetuned version was trained on largely positive compliance samples. The results match our observation with human validation and MLLM evaluation, that our method gets the highest negative score while having better or comparable positive and quality scores with other open source guidance methods.

Table 8: Evaluation of Different Methods using Our NegAwareQwen-32B

|  | Positive Score ($\uparrow$) | Negative Score ($\uparrow$) | Quality Score ($\uparrow$) |
|---|---|---|---|
| VSF Quality | 0.980 | 0.330 | **0.814** |
| VSF Strong | 0.870 | **0.415** | **0.814** |
| NASA | 0.950 | 0.224 | 0.727 |
| NAG Strong | 0.950 | 0.168 | 0.795 |
| NAG | **1.000** | 0.147 | 0.812 |
| GPT-4o | **0.978** | **0.619** | 0.812 |
| Nano Banana | 0.985 | 0.406 | **0.817** |

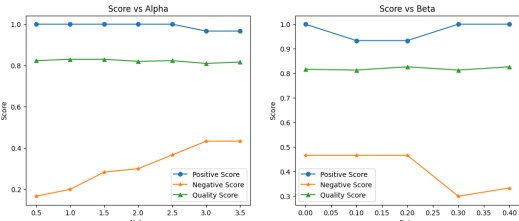

Figure 19: Relationships between metrics and $\alpha$ and $\beta$

## Q  ABLIATION STUDY ON $\alpha$ AND $\beta$

To study the effects of $\alpha$ and $\beta$ and hyperparameter sensitivity, we studied the effects of the two hyperparameters. We used 30 randomly selected prompts from the dataset and tested the effects of $\alpha$ and $\beta$ on the positive, negative, and quality scores. When testing the effects of $\alpha$, we set $\beta$ to 0, and when studying the effects of $\beta$, we set $\alpha$ to 3.5. All generations use the same seed. The images are evaluated using our NegAwareQwen-32B running for negative and quality score and used LLaMA for the positive score. Qualitative results of the effects of $\alpha$ and $\beta$ are shown in Figure 10 and quantitative results are shown in Figure 19. We can see that as $\alpha$ increases, negative scores increase while positive scores decrease. When $\beta$ increases, the negative score decreases while the positive score increases, which could be noise. In both cases, the quality scores only change slightly.

## R  FUTURE WORK

Future work may involve applying it to non-diffusion models (like Janus-4o (Chen et al., 2025b)) or models with complex text encoders (like Qwen-Image (Wu et al., 2025)), improving robustness through normalization and blending techniques similar to those employed by NAG, and optimizing computational efficiency by using a better attention implementation. Additionally, we observed some inaccuracies in MLLM judgment due to ambiguities or minimal differences in visual differences. Conducting a larger-scale human evaluation study would help mitigate inaccuracies observed in MLLM-based assessments. Investigating the attention maps and diffusion trajectories of our model could further elucidate the underlying mechanisms of VSF. Decoupling the attention, such that it calculates the positive and negative attention separately and then uses the ratio to extrapolate the output, might yield better quality in exchange for runtime. Or, adding a scaling factor to the positive prompt for better control.