# OpenReview forum: "VSF: Simple, Efficient, and Effective Negative Guidance in Few-Step Image Generation Models By Value Sign Flip"
_ICLR.cc/2026/Conference — ICLR 2026 Poster_

### Official Review · Reviewer_2X36 · 2025-10-31

**Soundness:** 3
**Presentation:** 3
**Contribution:** 3
**Rating:** 6
**Confidence:** 4

**Summary:**

The paper proposes Value Sign Flip (VSF), a lightweight, training-free negative-guidance method for few-step diffusion/flow models that flips the sign of negative-prompt value vectors inside attention. For cross-attention models VSF concatenates positive/negative keys/values and applies a \alpha scaling to the negative values; for MMDiT-style models (e.g., SD 3.5 Turbo, FLUX Schnell) it further duplicates the negative tokens and masks attention paths so only image -> negative interactions are affected, optionally adding a bias to stabilize quality. VSF adapts token-, layer- and step-wise, aiming to avoid oversaturation and “mixing” failures seen when forcing CFG or using fixed-strength attention methods like NASA and NAG. On a new negation benchmark (NegGenBench) built from challenging positive/negative prompt pairs, VSF improves negative adherence while keeping quality and positive adherence competitive, and runs in ~3 s with few steps; human and MLLM-based evaluations broadly agree with these trends.

**Strengths:**

- The proposed VSF is simple and plug-and-play. And it does not include many runtime overhead.
- It performs better than baseline methods on the collected benchmarks. It achieves better negative-prompt compliance at similar quality vs. NAG/NASA and even CFG (multi-step) in reported settings. The visualization results look visually good.
- Clear ablations/trade-offs (α, β, masking/duplication, Whole-Embedding Flip)

**Weaknesses:**

- Evaluation relies heavily on MLLM judges (LLaMA/Qwen variants) for pos/neg/quality scoring; such metrics can be biased or insensitive to artifacts, and the dataset/hyperparameters are author-curated.
- Some quality trade-offs remain at higher negative strength; masking/bias choices introduce extra knobs and implementation complexity in MMDiT stacks.

Writing:
-  Figure 3, 5, 6 is too small, could be scaled up for better organization.

**Questions:**

see weaknesses

---

> ### Author Response · Authors · 2025-11-21
>
> Thank you to the reviewer for the feedback and scoring.
> 1. We acknowledge the potential inaccuracy and bias of using MLLM as a judge, and thus, we already included human evaluation in the paper (Table 3). MLLM judges are currently a standard practice in recent T2I and negative-guidance literature, and our evaluation follows this widely adopted protocol.
> 2. We will open-source all of our code, including dataset generation, after publication. That means others can generate new prompts and try different hyperparameters to cross-examine our results. We already swept the hyperparameter for NAG and NASA for fairness, and showed that our method has a better trade-off.
> 3. Our method has a better trade-off than other methods (Figure 7). Our method can reach a higher negative score with less degraded positive/quality scores.
> 4. The attention bias in our method is optional and does not have a very strong effect on the final results (see Figure 7 Down). The implementation is also simple in many diffusion frameworks (see pseudo code in Listing 1).
> 5. We enlarged the figures.

---

### Official Review · Reviewer_qk8d · 2025-11-01

**Soundness:** 3
**Presentation:** 2
**Contribution:** 2
**Rating:** 4
**Confidence:** 4

**Summary:**

- Since previous negative prompts approach generally employ score matching, they are not compatible with few-step generation models.
- To address the problem, they simply flip the value in the attention layer, which results in effectively removing the undesired contents in the final images.

**Strengths:**

- The proposed method is simple but effective
- The proposed method can be applied to a few-step model.

**Weaknesses:**

- The proposed method is not novel. Manipulating attention has been employed for image editing with diffusion models and the flow-matching model. (e.g., [Attend-and-exit], [self-guidance], [BoxDiff])
- In Figure 5, VFS not only eliminates the undesired contents but also changes the other components. Specifically, in the starry night examples, the city has gone. Also, in Figure 5(right), the car is still reflected in the image with the proposed method.
- Figures should be well illustrated. ( font size of figure 3,5 is too small, and figure 6 is too small)

[Attend-and-exit]: Attention-Based Semantic Guidance for Text-to-Image Diffusion Models
[self-guidance]: Diffusion Self-Guidance for Controllable Image Generation
[BoxDiff]: Text-to-Image Synthesis with Training-Free Box-Constrained Diffusion

**Questions:**

- Compared to the previous works using attention manipulation, does the proposed method contain a specific technique or contribution for being compatible with few-step models?

---

> ### Author Response · Authors · 2025-11-21
>
> Thank you for the reviewer for the feedback and scoring.
> 1. Novelty relative to prior attention-manipulation work: We agree that attention editing appears in before work. Attention is one of the key elements in diffusion/flow-matching models, and thus, there would be many models focusing on that. The contribution is how negative guidance is made compatible with few-step DiT/MMDiT models. Our novelty is that we used attention as a way to introduce negative guidance in few-step models. VSF flips negative-token values inside the attention computation with a masking/duplication scheme designed for step-distilled architectures. It can be applied in a few-step model, compared to CFG, is that we do not do any extrapolation in the output space but in the attention space. This produces stable, token-adaptive negative guidance in a single pass. It shares similarity with previous work in that we both manipulate attentions, but we used it in a way to cancel unwanted elements instead of changing elements' properties in the image (Self-Guidance), setting specific object locations (BoxDiff), and prompt following by forcing the model to attend to more tokens (Attend-and-Excite). Additionally, our method engineered the input to attention (values) and not the attention map directly. We have added these related works to the paper and clarified our differences (line 196).
> 2. Removing “Vincent van Gogh style’’ removes Van-Gogh-specific town elements: We argue that this is intended. The example demonstrates not just style avoidance, but also semantic concept avoidance. A painting of “Starry Night” without the “Vincent van Gogh style”, will produce a generic starry night image, which does not necessarily contain the town element. Here, we are not producing “THE Starry Night” painting without Vincent van Gogh's style, but a generic Starry Night image, with any association with “Vincent van Gogh style” removed.
> In the car example, the positive and negative prompts both contain “car”. Our research question here is that if we can create abstract art by using conflict prompts. Under such conflicting prompts, the expected behavior is not full removal but a reduction of the semantic strength of the concept. Therefore, the resulting “partial presence” of the car reflects the intended behavior of concept weakening rather than a failure case. We will clarify this in the caption.
> 3. Figure readability: We have enlarged the fonts and adjusted layouts in Figs. 3, 5, and 6 for improved clarity.
> 4. Compatibility: Our contribution is that we flipped the sign of the negative prompt, which introduced a new way to perform negative guidance in few-step models. Compared with previous attention-manipulation methods, our method is targeted at the task of negative guidance, instead of general image editing. Our task on our dataset NegGenBench is also much harder as it involves de-assiostaion of closely related objects (e.g., bike and wheels) instead of changing the properties and locations of the objects in the image. Compared with other negative guidance methods in few-step models (NASA and NAG), our method is simpler yet more effective. Compared with traditional negative guidance methods (CFG), our sign-flip is performed on the attention level and not the final output, making it compatible with few-step models.

---

### Official Review · Reviewer_emZq · 2025-11-02

**Soundness:** 3
**Presentation:** 3
**Contribution:** 3
**Rating:** 8
**Confidence:** 4

**Summary:**

This paper proposes Value Sign Flip (VSF): a negative guidance.

Negative guidance is useful in generative pipelines: progressively adjusting the results by removing something from the image.

Problem statement:
* CFG does not work in few-step configurations.
* Negative Steer Away Attention (NASA) is currently limited to cross-attention models.
* Normalized Attention Guidance (NAG) primarily targets quality control rather than avoiding negative prompts.
* NASA and NAG subtract negative attention.
* They do not generalize to various timesteps, layers, or image regions.

Method (VSF)
* (Cross-attention-based models) Flipping the sign of negative prompt values within the attention calculation.
* (DiT models) Duplicating negative prompts, one remains unflipped, another is flipped.

Advantages
* VSF removes some concepts from generated images: wheels from bicycle, hands from clock, etc.
    * maintaining image quality and adherence to the positive prompts
* VSF works in few-step (1~8 steps) diffusion and flow matching models.
* VSF generalizes to SD 3.5 turbo, FLUX schunell, and Wan.
* VSF is computationally cheap.

New dataset: NegGenBench
* Positive-negative prompt pairs from ChatGPT o3.
* Negative prompts are core components from positive prompt; it is intentionally challenging.

Evaluation: fine-tune a VLM (Qwen) for measuring faithfulness to negative prompts

**Strengths:**

Originality:
1. The proposed method is new.

Quality:
1. The related work section covers relevant literature: CFG, Negative Guidance, and Few-step generators
2. The competitors are aggressively chosen, even Nano Banana.
3. Discussion is thorough
    1. trade-off between positive and negative prompts
    2. trade-off between quality and negative prompts
    3. attention maps
    4. ablation study

Clarity:
1. The explanations are kind to the readers, step-by-step from NASA to the proposed method.

Significance:
1. The method is simple and effective.
    1. simple: concatenate the values and keys of the positive and negative prompts, then flip the sign of the negative prompt values
    2. effective: Table 2

**Weaknesses:**

minor
1. Please properly use \citet and \citep
2. Fonts are too small in the figures.
3. Is “unbrulla” a typo? or is there a message?

**Questions:**

1. Why does NASA have points with negative score below 50 only in Figure 6?

---

> ### Author Response · Authors · 2025-11-21
>
> Thank you to the reviewer for the feedback and scoring.
> We have fixed the citation typos and enlarged the figures."unbrulla" is a typo and we have fixed it. Thank you for catching that. About the question, NASA cannot reach a negative score higher than 50 even when pushing to extremes, which is why its negative score is always below 50.

---

### Official Review · Reviewer_DjfS · 2025-11-05

**Soundness:** 3
**Presentation:** 2
**Contribution:** 3
**Rating:** 6
**Confidence:** 4

**Summary:**

This paper proposes a simple approach for negative guidance of the text-to-image (T2I) models. Current approaches flip the sign of the attention output for this, but this ends up on applying the same scale of the negative guidance across different areas of the image, and all different layers of the model. Instead, they use the attention map to calculate a per-token wieght for this negative guidance. They first develop this for approaches that use cross-attention mechanism (like latent diffuoin architecture), and then propose a new mechanism to adapt this to multimodal diffusion transformer (MMDiT)-based models like SD3, and others.

**Strengths:**

The idea of this paper is simple, but I like how they have distilled knowledge from the literature, and based on that—as well as their solid understanding of the attention mechanism—they have proposed this simple idea.

**Weaknesses:**

1) There are some grammatical errors and confusing parts in the paper that need to be addressed:
- Should $x_{t-1}$ be $x_{t+1}$ in Eq. (1)?
- line 166: *"The method NASA applies the guidance in intermediate states instead of the predicted noise or velocity."* — this statement is somewhat ambiguous.
- line 188: "*However, it also limits the model’s ability to follow negative prompt guidance if the constraint is set to be too tight ...*" — this sentence could be improved for clarity and readability.

2) The concepts used to illustrate the issue with generation quality in Figure 2, under the presence of negative and positive prompts, are not optimal. Since winter and snow are strongly related concepts, they may be entangled in the diffusion model’s learned distribution. Using one as a positive and the other as a negative prompt may not clearly demonstrate the intended issue with current models, as the observed effect could stem from this conceptual dependence rather than the model’s capability to interpret negative guidance.

3) This paper mentions that "*rendering prompts containing negations ineffectively or made the negative prompt appears even more (e.g., a prompt like “a scientist who is not wearing glasses” will often generate a scientist with glasses—sometimes even more frequently than a simple prompt like “a scientist”).*".
There are some fairness-oriented approaches, such as ITI-Gen [1] and FairQueue [2], that discuss related issues. They point out that this cannot be addressed using **hard prompts**, but can be mitigated through prompt learning. While I understand that your setup is different, discussing the similarities and differences between your approach and these works could strengthen the paper.

4. I believe the task of **negative guidance** can be viewed as a special case of **image editing**, where the prompt explicitly describes the removal of a concept while no input image is provided. From this perspective, using Qwen-Image as an external baseline (Table 1) is an interesting choice. Given that it can edit images while keeping other regions intact, it could serve as an informative upper bound for the negative guidance task. The performance gap could also highlight directions for future research. However, the details of how Qwen-Image is used, including the experimental setup, should be included in the main paper (at least at a high level).

$ $

*References:*

[1] ITI-GEN: Inclusive Text-to-Image Generation, ICCV'23

[2] FairQueue: Rethinking Prompt Learning for Fair Text-to-Image Generation, NeurIPS'24

**Questions:**

Please check the weaknesses.

---

> ### Author Response · Authors · 2025-11-21
>
> Thank you to the reviewer for the feedback and scoring.
> 1. We have fixed Eq. 1.
> 2. We have changed “The method NASA applies the guidance in intermediate states instead of the predicted noise or velocity” to “NASA applies the guidance in intermediate states (attention outputs) instead of the final predicted noise or velocity.”
> We changed “However, it also limits the model’s ability to follow negative prompt guidance if the constraint is set to be too tight ...” to “However, if the constraint is set to be too tight (i.e., high $\alpha$ and low $\tau$), it might also limit the model's ability to follow negative prompt. “
> 3. In Figure 2, we have changed the prompt pairs to a winter landscape of Canada with a negative prompt of “lake”. These two concepts are bound, but not as strong as winter vs. snow.
> 4. We acknowledge the relation between our work and previous debias work. We have added those in the related works (line 196).
> 5. We want to clarify that we did not use Qwen-Image as an Image Editing baseline; we used Flux Kontext as the image editing baseline, as it was released after we finished the paper. Qwen-Image is used as a regular generation baseline (not editing). We included more detail about the Flux Kontext (Line 322)  image editing in the main text. Interestingly, it doesn’t really outperform our method.

---

### Author Response · Authors · 2025-12-02

Final Author Comment:
We want to thank all reviewers for giving their valuable comments, and we also want to thank the AC for taking extra time to read each comment and rebuttal, given the special circumstances this year.

Some reviewers pointed out some missed prior work; we have added them into the related work section, including debias work (from reviewer DjfS) and attention manipulation work (from reviewer qk8d).

Reviewer qk8d raised some concerns about the effectiveness and side effects of our work, specifically, why VSF removed the town when generating a starry night image without Van Gogh style. We want to clarify that it is intended as the town is a signature part of Van Gogh’s Starry Night. They also raised questions about whether the car is completely canceled in the image with cars as a negative prompt. We want to clarify that it is also intended as we want to semi-cancel the car by putting the car in both positive and negative prompts to create a semi-abstract style.

Reviewer 2X36 also questioned about the reliability of MLLM as judge; we have verified using human evaluation in the original paper.

There are some other minor concerns regarding figure size and typos; we have enlarged the figure and fixed typos. We also clarified wording that could be confusing. We also added more context for the image editing baseline in the main text.

---

> ### Author Response · Authors · 2025-12-04
>
> One more further work direction is to use image as negative prompt, as our guidance happens in embedding space not output space. We can directly replace the textual negative prompt with a CLIP or VAE image embedding. Similar to [1].
> [1] Training-Free Safe Denoisers for Safe Use of Diffusion Models

---

### Meta-Review · Area_Chair_w6tj · 2026-01-05

**Summary:**

This paper received mixed reviews. On the positive side, the method here is simple to implement and adds little computational overhead, improving negative prompt adherence. The authors show attention manipulation for both cross-attention and self-attention in MMDiT. On the other hand, the novelty is limited, as manipulating attention is a common approach for image editing and generation. Also, the method here is applied to few-steps setting, but no specific design is intended for few-step generation.

The AC recommends accepting this paper for its technical validity and practical usefulness.

**Reviewer Concerns:**

Concerns that were addressed by the rebuttal:
- missing related work on fairness-oriented image generation (reviewer DjfS). The authors have discussed these suggested papers in the revised version.
- missing comparison to image editing method (reviewer DjfS). The authors have included FLUX image editing as a baseline.
- Evaluation heavily relies on MMLM, which may not be trustworthy (reviewer 2X36). The authors have included human evaluation in the revised version.

Outstanding concerns:
- Limited novelty, as manipulating attention is a common method for image editing/generation. (reviewer qk8d)

**Reviewer Scores:**

The reviewers are likely to maintain their initial ratings (4,6,6,8).

---

### Decision · Program_Chairs · 2026-01-26

Accept (Poster)